# HOLON-OF-THOUGHT: IMPROVING ROBUSTNESS IN LARGE LANGUAGE MODELS VIA STRUCTURED FRAMEWORK

## ABSTRACT

Large Language Models (LLMs) excel in language comprehension and generation tasks but frequently face challenges in scenarios demanding rigorous logical reasoning or strict adherence to problem conditions. In such reasoning, errors propagate through intermediate steps, hallucinatory outputs violate key problem conditions, and complex problems are often handled in a simplistic, chain-like manner. We propose Holon-of-Thought (**HoT**), a structured reasoning framework. **HoT** explicitly extracts problem conditions and enforces their adherence. It dynamically decomposes complex problems into verifiable subtasks and solves them through a four-stage pipeline: condition extraction, path exploration, adaptive decomposition, and aggregation. The experimental results show that **HoT** improves the accuracy of the inference and enhances the robustness. This establishes a new paradigm for reliable LLM-based reasoning in mathematics and logic.

## 1 INTRODUCTION

The rapid advancement of Large Language Models (LLMs) has driven a transformative shift in artificial intelligence. LLMs now demonstrate strong capabilities in natural language processing, knowledge retrieval, and complex task automation Brown et al. (2020); Naveed et al. (2023). However, despite their impressive breadth of capabilities, LLMs often falter when tasked with rigorous reasoning Wei et al. (2022); Wang et al. (2024a).

This brittleness stems from their training paradigm: LLMs learn statistical correlations rather than formal logic, making them prone to errors when faced with problems requiring deductive certainty or strict condition satisfaction Brown et al. (2020); Bender et al. (2021). This shortcoming is especially evident in scenarios requiring strict adherence to logical conditions or precision-oriented decision-making. For example, in mathematical proof generation or engineering design verification, where a single misstep invalidates the entire solution, LLMs often produce outputs that are locally plausible but globally inconsistent First et al. (2023); Lu et al. (2024).

In addition, their output often contains hallucinations, which are associated with the neglect of problem conditions or facts Ji et al. (2023); Huang et al. (2025); Zhang et al. (2025). These errors reduce the reliability of LLMs in high-stakes applications, where factual inaccuracies can lead to severe consequences Thirunavukarasu et al. (2023); Dahl et al. (2024); Niu et al. (2024). Hallucinations typically arise when models fill knowledge gaps with statistically plausible but unsubstantiated content Huang et al. (2025); Tonmoy et al. (2024).

To address these issues, a variety of reasoning-enhancement strategies have been proposed. Among them, Chain-of-Thought (CoT) Wei et al. (2022) prompting has emerged as a widely used approach that encourages models to explicitly enumerate intermediate steps during reasoning. By externalizing the reasoning process, CoT provides a window into the model's "thinking", aiding both performance and interpretability Kojima et al. (2022). Although CoT improves performance on many multistep problems, it still exhibits brittleness: it can overlook hard conditions, generate invalid intermediate steps Wang et al. (2023); Arcuschin et al. (2025). This underscores the need for more structured, condition-aware reasoning frameworks to achieve robust and interpretable reasoning.

For the sake of robust and interpretable reasoning, we propose Holon-of-Thought (**HoT**). **HoT** is a structured reasoning framework with a four-stage pipeline of condition extraction, path exploration, adaptive decomposition, and aggregation, explicitly enforcing condition adherence and dynamically decomposing problems into verifiable subtasks. The focus on exploration and condition satisfaction reflects the strategies used in combinatorial optimization and automated planning.

This approach draws on classical condition satisfaction systems and modern adaptive computation techniques, balancing thoroughness with efficiency Graves (2017). **HoT**'s architecture is designed to be model-agnostic, operating purely through prompt engineering or lightweight API calls, ensuring wide applicability without retraining overhead.

**HoT** innovates by explicitly extracting and enforcing both explicit and implicit problem conditions, generating and scoring multiple high-level solution paths to select the optimal one, adaptively decomposing complex problems into isolated verifiable subtasks based on complexity, and aggregating sub-solutions while ensuring global condition adherence. These innovations offer three key advantages: (1) robust reasoning through explicit condition prioritization, reducing error propagation; (2) improved interpretability through structured, auditable reasoning traces; and (3) computational efficiency through selective reasoning, enabled by pruning—generating multiple candidate methods and retaining only the optimal one, thus avoiding wasteful exploration of dead ends.

Our work underscores that achieving robust reasoning in LLMs requires condition-aware architectural designs that prioritize structure, precision, and verifiability. **HoT** exemplifies this principle by promoting a disciplined approach to reasoning. The structured methodology enables LLMs to reason more conservatively and avoid compounding errors, especially in tasks where correctness is tightly coupled with condition satisfaction. By combining selective exploration with rigorous synthesis, **HoT** provides a scalable blueprint for deploying LLMs in engineering applications where reliability and interpretability are paramount.

## 2 RELATED WORK

Prompt-based reasoning aims to unlock the complex capabilities of LLMs without expensive fine-tuning. The paradigm was pioneered by CoT prompting, which generates intermediate steps to trace a sequential reasoning process Wei et al. (2022). This concept was extended by methods like Tree of Thoughts Yao et al. (2023a) and Graph of Thoughts Besta et al. (2024), which explore non-linear reasoning paths using more expressive tree and graph structures, respectively, to manage complex problem solving. In contrast, **HoT** differentiates itself by integrating upfront condition extraction and adaptive decomposition into its path exploration.

The field has since expanded rapidly, with research exploring numerous avenues to enhance LLM's reasoning. Many efforts have focused on iterative refinement, where models critique and improve their own outputs, such as Self-Refine Madaan et al. (2023), Step-Back Zheng et al. (2024) and System 2 Attention Weston & Sukhbaatar (2023). Other approaches incorporate external formalisms to add rigor. For example, Logical Thoughts Zhao et al. (2023) integrates symbolic logic, while other methods use structured formats like symbolic expressions, tables, or executable code to offload computation and enforce syntactic correctness Hu et al. (2024); Wang et al. (2024b); Puerto et al. (2024); Gao et al. (2023). These methods demonstrate the diverse strategies being investigated to make LLM's reasoning more powerful and reliable. In contrast to these iterative or formalism-based techniques, **HoT** focuses on grounding reasoning in extracted conditions and dynamically adapting the problem structure. **HoT** offers a framework that complements these methods by emphasizing condition enforcement without repeated critiques or extensive external tools.

A significant challenge is ensuring the robustness of generated reasoning, as standard CoT is often susceptible to process errors or hallucinations. To address this, Self-Consistency Wang et al. (2023) and LLM-Blender Jiang et al. (2023) mitigate errors via multitrajectory consensus, though this brute-force approach incurs high computational costs. EchoPrompt Mekala et al. (2024) seeks efficiency by distilling divergent rationales into a unified path but risks reinforcing errors if initial paths are flawed. This reveals a core tension: aggregating diverse paths for robustness can be either computationally expensive or risk converging on an incorrect solution. Some technologies avoid this problem by using prompt words. For example, "specify constraints pattern" Moundas et al. (2024) was proposed to process constraints and reduce noise interference, but this method only op-

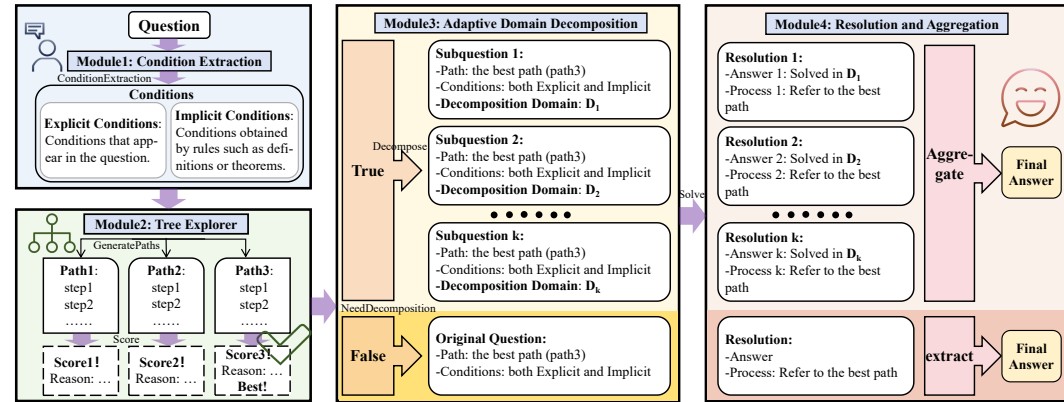

Figure 1: **HoT** Framework. It includes four modules: Condition Extraction, Tree Explorer, Adaptive Domain Decomposition, and Resolution and Aggregation. Condition Extraction identifies both explicit and implicit conditions from the original problem. Tree Explorer generates three potential solution paths and selects the one with the highest score as the basis for subsequent reasoning. Adaptive Domain Decomposition determines whether the problem should be decomposed, based on its complexity and the extracted conditions. If decomposition is necessary (as illustrated in the top half of the diagram), the problem is split into sub-problems. Resolution and Aggregation then solves each sub-problem individually and combines their results to generate the final answer. If decomposition is not needed (as shown in the bottom half), Resolution and Aggregation directly solves the original problem and outputs the final result.

erates at the level of data annotation and ignores the crucial role of implicit conditions in reasoning. Contrastive Chain-of-Thought Prompting Chia et al. (2023) provides both positive and negative exemplar reasoning chains to guide the model away from common mistakes, improving reasoning quality in a structured way. Contrastive Denoising with Noisy Chain-of-Thought Zhou et al. (2024) constructs noisy-rationale scenarios and learns to denoise rationales by contrasting noisy and clean ones. Chain-of-Defensive-Thought Wang et al. (2025) uses structured, defensive reasoning exemplars to enhance robustness. These methods improve robustness but often at the cost of flexibility, computational efficiency, or general applicability.

Another critical research direction tackles the rigidity of linear reasoning through two intertwined strategies: problem decomposition and process adaptability. For decomposition, methods like Thread-of-Thought (ThoT) Zhou et al. (2023) segment complex inputs, while architectural innovations like Layer-of-Thoughts (LoT) Fungwacharakorn et al. (2024) impose predefined hierarchies. However, their static nature limits effectiveness: ThoT relies on brittle fixed segmentation heuristics, and LoT's rigid schemas may misalign with task logic. Other methods focus more on process adaptability. For instance, Buffer of Thoughts (BoT) Yang et al. (2024) uses memory to reuse prior reasoning segments but prioritizes past solutions over adapting structure for novel tasks. Recursive Decomposition of Logical Thought (RDoLT) Qasim et al. (2025) recursively decomposes tasks with scoring and knowledge propagation but faces scalability limits due to computational overhead. Instance-Adaptive Prompting (IAP) Yuan et al. (2024) adapts the prompt based on input complexity, but this only tunes the starting point, leaving the subsequent single-path reasoning flow unchanged. This highlights the need for a framework that can dynamically structure its reasoning process, flexibly decomposing the problem as needed, and does not require too much computational resources.

## 3 METHODOLOGY

We design the **HoT** framework to significantly enhance the reasoning robustness of LLMs by explicitly extracting problem conditions and systematically exploring the solution space. **HoT** guides the model toward logically grounded reasoning paths and mitigates error propagation across complex problem decompositions. As shown in Figure1, the **HoT** framework contains four modules: Condition Extraction, Tree Explorer, Adaptive Domain Decomposition, Resolution and Aggregation.

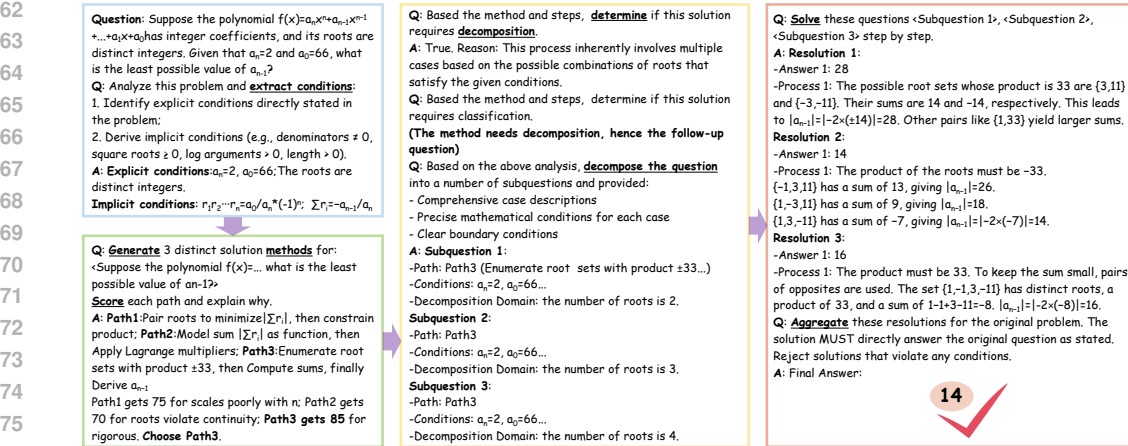

Figure 2: Example of **HoT** Prompting in Solving a Complex Math Problem. This figure corresponds one-to-one with Figure 1, illustrating the practical execution of each module and flow in the **HoT** framework. It demonstrates the application of the **HoT** framework to a polynomial root problem. It shows the decomposition of the original problem into subproblems based on the number of roots, the resolution of each subproblem using the selected optimal path (Path 3), and the aggregation of results to derive the final answer. This example validates HoT's effectiveness in handling multi-case mathematical reasoning through structured decomposition and verification.

## 3.1 CONDITION EXTRACTION

**HoT** utilizes Condition Extractor module to transform raw question statements into a structured representation by identifying explicit and implicit conditions. This module acts as the "grounding phase," forcing the LLM to explicitly articulate the rules and boundaries governing the problem before attempting solutions, addressing a common failure mode where models overlook implicit conditions. It structures the conditions and transforms the fuzzy input into a computable framework, helping prevent failure in subsequent steps due to missing information. Its output acts as a shared, immutable condition set referenced throughout the **HoT** pipeline.

In our definition, conditions $C$ are divided into explicit conditions $\mathcal{C}_e$ and implicit conditions $\mathcal{C}_i$. $\mathcal{C}_e$ are directly parsed from mathematical formulations or logical statements, while $\mathcal{C}_i$ are derivable through mathematical or contextual rules or common sense knowledge relevant to the domain (e.g., "ages must be positive integers", "a triangle's angles sum to 180 degrees").

Given a question $Q$, we get $\mathcal{C}_e$ and $\mathcal{C}_i$:

$$\text{ConditionExtraction}_{\text{LLM}}(Q) \rightarrow (\mathcal{C}_e, \mathcal{C}_i) \tag{1}$$

For example, given the question $Q$: "What is the sum of the three digit cubes that are the cubes of either squares or cubes?" We can distill the explicit conditions, "The cubes must be of numbers that are either squares or cubes themselves", and the implicit conditions, "The three-digit cubes range from 100 to 999, so the cube roots must be integers between 5 and 9 inclusive, because $4^3$=64 (too small) and $10^3$=1000 (too large)". The distillation of these conditions will directly guide subsequent work.

## 3.2 TREE EXPLORER

**HoT** uses Tree Explorer module to determine a path to solve the question. Tree Explorer module firstly generates a set of viable solution paths, $\mathcal{P} = \{\pi_1, \pi_2, ..., \pi_N\}$:

$$\text{GeneratePaths}_{\text{LLM}}(Q, \mathcal{C}_e, \mathcal{C}_i) \rightarrow \mathcal{P} \tag{2}$$

Each path $\pi_i$ represents a high-level strategy, detailing the proposed method and concrete implementation steps. This is not a search through intermediate steps, but a generation of complete, end-to-end strategies. By generating multiple paths, the model explores a diverse range of question-solving paradigms before committing to a single approach.

Once multiple paths are proposed, the next phase involves a rigorous evaluation to select the most promising one.

### 3.2.1 PATH SCORING

Each path $\pi_i$ is systematically evaluated by LLM based on a predefined set of criteria, such as anticipated accuracy, operational feasibility, and computational complexity. The model assigns a Heuristic score, $s_i \in [0, 100]$ to each path, resulting in a set of scored tuples:

$$\mathcal{P}_{\text{scored}} = \{(\pi_i, s_i) \mid s_i = \text{Score}_{\text{LLM}}(\pi_i, Q, \mathcal{C}_e, \mathcal{C}_i)\} \tag{3}$$

### 3.2.2 OPTIMAL PATH SELECTION

The path with the highest score is selected as the optimal strategy, $\pi^*$.

$$\pi^* = \underset{\pi_i \in \mathcal{P}}{\arg\max}\, s_i \tag{4}$$

Optimal Path Selection is a decisive step that dictates the entire subsequent execution flow.

Tree Explorer introduces a critical self-reflection step, enabling the model to deliberate on the quality of its own plans before execution. Also, Tree Explorer provides a classification basis for Adaptive Domain Decomposition.

## 3.3 ADAPTIVE DOMAIN DECOMPOSITION

Adaptive Domain Decomposition module determines the final execution strategy by assessing the question's complexity relative to the chosen path $\pi^*$. It decides whether a divide-and-conquer approach is necessary.

First, the framework performs a binary classification to determine if the question domain should be partitioned:

$$
\begin{aligned}
DecomposeFlag &= \text{NeedDecomposition}_{\text{LLM}}(\pi^*) \\
&\in \{\text{True}, \text{False}\}
\end{aligned}
\tag{5}
$$

When $DecomposeFlag$ is True, **HoT** partitions the problem into logically isolated subproblems $\mathcal{Q}_{sub} = \{Q_1, Q_2, ..., Q_k\}$. This decomposition strategically splits the feasible domain defined by $\mathcal{C}_e$ and $\mathcal{C}_i$, aligning subproblem boundaries with critical decision points in $\pi^*$. Each subproblem inherits relevant conditions and operates in semantically isolated containers—ensuring errors in one subdomain cannot propagate to others. This approach transforms complex combinatorial, multi-case, or recursive problems into parallelizable verification tasks while maintaining strict condition adherence.

$$\mathcal{Q}_{sub} = \text{Decompose}_{\text{LLM}}(Q, \pi^*, \mathcal{C}_e, \mathcal{C}_i) \tag{6}$$

When $DecomposeFlag$ is False, the problem does not need to be decomposed and can be solved directly into the next Module (see Direct Resolution for Resolution and Aggregation).

## 3.4 RESOLUTION AND AGGREGATION

Resolution and Aggregation module executes the plan established in Adaptive Domain Decomposition module. The reasoning process follows one of two pathways based on the outcome of the adaptive decomposition. This bifurcation ensures computational efficiency for simple problems while maintaining rigorous error isolation for complex ones, adapting dynamically to the problem's needs.

- Direct Resolution: when $DecomposeFlag$ is False, the question is considered monolithic. The model applies the chosen strategy $\pi^*$ to solve the original question Q in a single, direct pass while ensuring the solution satisfies $\mathcal{C}_e$ and $\mathcal{C}_i$. This path is typical for problems with short reasoning chains or those where decomposition would introduce unnecessary overhead (e.g., single-step arithmetic, straightforward logical inferences). The final solution is obtained as:

$$A = \text{Solve}_{\text{LLM}}(Q, \pi^*, \mathcal{C}_e, \mathcal{C}_i) \tag{7}$$

- Hierarchical Resolution: when $DecomposeFlag$ is True, the model engages in a multi-step hierarchical process. First, it independently resolves each subquestion $Q_i \in \mathcal{Q}_{sub}$ using the logic of $\pi^*$, yielding a set of sub-answers $\mathcal{A}_{sub} = \{A_1, A_2, ..., A_k\}$, while ensuring all solutions strictly adhere to $\mathcal{C}_e$ and $\mathcal{C}_i$. Each subquestion is solved in isolation, preventing error propagation between subproblems. For instance, in a problem requiring case analysis (e.g., "Solve for x where x² + bx + c = 0, considering discriminant cases"), each subproblem corresponds to a distinct case (D>0, D=0, D<0), and solving one case incorrectly does not affect others. For the answer to each subquestion, there is the following relationship equation:

$$A_i = \text{Solve}_{\text{LLM}}(Q_i, \pi^*, \mathcal{C}_e, \mathcal{C}_i), \forall Q_i \in Q_{sub} \tag{8}$$

  Next, these partial solutions are synthesized. The model aggregates the information from $\mathcal{A}_{sub}$ to construct a single, coherent, and comprehensive final solution and confirm that the aggregated answer satisfies all conditions, $\mathcal{C}_e$ and $\mathcal{C}_i$, directly answers the original question Q. Aggregation rules are problem-specific. For summation problems, it might involve simple addition; for case analysis, logical combination; for condition satisfaction, intersection of valid solutions.

$$A = \text{Aggregate}_{\text{LLM}}(\mathcal{A}_{sub}, Q, \mathcal{C}_e, \mathcal{C}_i) \tag{9}$$

Figure1 illustrates this dynamic pipeline, highlighting how conditional execution optimizes the trade-off between thoroughness (for complex tasks) and efficiency (for simpler ones). **Algorithm**1 shows the **HoT** reasoning process in a formal description. Figure2 illustrates an example of solving a complex math problem using **HoT** framework. Here, since $DecomposeFlag$ is True, the problem undergoes decomposition and aggregation, and the correct answer is obtained.

This conditional execution allows **HoT** to dynamically adapt its strategy, applying a more robust, multi-step reasoning process only when necessary, thereby optimizing for both accuracy and efficiency. Finally, the resulting answer $A$ is formatted for the end-user.

## 4 EXPERIMENTS

To rigorously evaluate the proposed **HoT**, we conducted a comprehensive set of experiments designed to assess its performance, generalizability, robustness, and the contribution of its core components. Our evaluation demonstrates that **HoT** achieves superior accuracy on a diverse suite of mathematical (GSM8K, ASDiv, SVAMP) and logical (OpenBookQA, Strategy) reasoning benchmarks. We further show that these performance gains are model-agnostic, enhancing the capabilities of multiple underlying LLMs. Critically, through quantitative stability metrics, we found that **HoT** not only provides more accurate results but does so with significantly greater consistency and lower variance than baseline methods. Finally, a detailed ablation study confirmed that each module of **HoT** is integral to its success, with its structured approach of identifying conditions, decomposing problems, and exploring solution paths being fundamental to its effectiveness.

### 4.1 EXPERIMENTAL SETUP

#### 4.1.1 DATASETS.

We evaluate **HoT** with two types of datasets, Math: GSM8K Cobbe et al. (2021), ASDiv Miao et al. (2021), SVAMP Patel et al. (2021), and Logic: OpenBookQA Mihaylov et al. (2018), StrategyQA Geva et al. (2021). These datasets share common characteristics: a certain depth of thought and the need to synthesize knowledge and reasoning. For each dataset we take the first 200 examples of the test set.

#### 4.1.2 MODELS.

We use Qwen2.5:7b-instruct Qwen et al. (2025) as the backbone model for our main experiments due to its superior semantic comprehension and execution capabilities. We also use DeepSeek-V3 DeepSeek-AI et al. (2025) to test the effectiveness of **HoT** and complete robustness testing via API. All the experiments on Qwen2.5:7b-instruct are run on an 1x NVIDIA A100 GPU server. The temperature hyperparameter T of models is set to 0.3.

### 4.1.3 BASELINES.

Our baselines include: CoT Wei et al. (2022), Random-CoT Fu et al. (2022), CoT with Self-Consistency (CoT-SC) Wang et al. (2023), ReAct Yao et al. (2023b), instance-adaptive prompting strategy (IAP) Yuan et al. (2024) and AoT Teng et al. (2025). For CoT-SC, we set the number of paths n = 5. For ReAct, we designed similarly styled prompts for each dataset as examples. For IAP, we adopt the Majority Vote strategy as our approach. Accuracy is the average value of the results of 3 runs, and detailed reproduction settings are provided in Appendix A.6.

### 4.1.4 METRICS.

We adopt both standard and newly designed metrics to evaluate different reasoning methods. In the main experiments and model comparison studies, we report the average accuracy over three independent runs. For robustness testing, we design two complementary metrics based on repeated testing: (1) **Total Variance (TV)**, and (2) **Instance Variance Mean (IVM)**. These metrics are defined in a general form to allow application across various experimental settings. Then we introduce how to obtain the two metrics.

Let each method be tested over $M$ independent runs. Each run consists of $N$ problems. The total runs are grouped into $B$ problem blocks, and each problem block contains $N$ problems. Each problem block is evaluated $R$ times. Thus, the total number of runs is $M = R \times B$.

Let $A_b^{(r)}$ denote the accuracy of the model on the $b$-th problem block in its $r$-th repetition, where $b \in \{1, 2, \ldots, B\}$, and $r \in \{1, 2, \ldots, R\}$. The mapping from the pair $(b, r)$ to the global run index $i$ is given by $i = (b-1) \times R + r$.

Let $A^{(i)}$ denote the accuracy of the $i$-th run (where $i \in \{1, 2, \ldots, M\}$ and the mapping from $(b, r)$ to $i$ is mentioned above). Then:

$$\text{TV} = \text{Var}(A^{(1)}, A^{(2)}, \ldots, A^{(M)})$$

For each block $b$, we compute the variance of the accuracy values across the $R$ repetitions:

$$V_b = \text{Var}(A_b^{(1)}, A_b^{(2)}, \ldots, A_b^{(R)}).$$

IVM is then defined as the average variance across all $B$ problem blocks:

$$\text{IVM} = \frac{1}{B} \sum_{b=1}^{B} V_b$$

## 4.2 EXPERIMENTAL RESULTS AND ANALYSIS

### 4.2.1 MAIN RESULTS.

As shown in Table1, **HoT** achieves excellent performance across all datasets. On SVAMP, it achieves 91.6% accuracy (+0.4% over AoT); on OpenBookQA, it reaches 91.5% accuracy (+3.9% over CoT-SC). **HoT \*** is derived from voting among CoT, HoT, and HoT without Tree Explorer (Module 2), with **HoT**'s answer as the tiebreaker. **HoT \*** further improves performance: 92.2% on GSM8K (+2.0% over AoT) and 92.2% on average (+1.2% over **HoT**). Result show that **HoT** is able to improve the accuracy of LLM reasoning whether solving mathematical or logical problems. With more computational resources, **HoT** can continue to improve LLMs' inference ability.

### 4.2.2 REASONING MODELS COMPARISON RESULTS.

We evaluate the effectiveness of the proposed **HoT** framework across two representative reasoning benchmarks (GSM8K and OpenBookQA) , using two LLMs with different parameter scales: Qwen2.5:7b-Instruct and DeepSeek-V3. Table2 shows that, **HoT** consistently improves performance across datasets. For Qwen2.5:7b-instruct, **HoT** boosts accuracy from 81.6% to 86.3% on GSM8K (+4.7%) and from 83.9% to 91.5% on OpenBookQA (+7.6%). For DeepSeek-V3, performance improves from 91.3% to 97.0% on GSM8K (+5.7%) and from 94.5% to 96.7% on Open-BookQA (+2.2%). The larger relative gains for Qwen2.5:7b-instruct suggest that **HoT** is particularly

| Methods | GSM8K | ASDiv | SVAMP | OpenBookQA | StrategyQA | Avg. |
|---|---|---|---|---|---|---|
| CoT | 81.8 | 93.2 | 86.1 | 83.9 | 84.8 | 86.0 |
| Random-CoT | 82.6 | 94.1 | 88.3 | 82.5 | 87.3 | 87.0 |
| CoT-SC (n=5) | 85.1 | 94.4 | 90.8 | 87.6 | 88.0 | 89.2 |
| ReAct | 84.1 | 94.7 | 90.9 | 87.1 | 85.1 | 88.4 |
| IAP | 80.5 | 91.1 | 87.2 | 75.3 | 88.8 | 84.6 |
| AoT | 90.2 | 97.2 | 91.2 | 84.9 | 86.5 | 90.0 |
| **HoT** (Ours) | 86.3 | 97.2 | 91.6 | **91.5** | 88.1 | 91.0 |
| **HoT \*** (Ours) | **92.2** | **97.7** | **93.8** | 88.3 | **89.3** | **92.2** |

Table 1: Performance Comparison of Reasoning Methods Across Datasets (%). This table compares the accuracy of **HoT** and **HoT \*** with baseline methods (CoT, Random-CoT, CoT-SC, ReAct, IAP) on five math and logic reasoning datasets. The experiment was conducted on the Qwen2.5:7b-instruct model. Results show **HoT** achieves a high average accuracy (91.0%) and **HoT \*** further sets a new state-of-the-art (92.2%), confirming their superiority in mathematical and logical reasoning tasks.

| Methods | | GSM8K | OpenBookQA | Avg. |
|---|---|---|---|---|
| Qwen2.5:7b-instruct | CoT | 81.6 | 83.9 | 82.8 |
| | **HoT** | 86.3 | 91.5 | 88.9 |
| DeepSeek-V3 | CoT | 91.3 | 94.5 | 92.9 |
| | **HoT** | **97.0** | **96.7** | **96.9** |

Table 2: **HoT**'s Performance Across Different LLM Backbones (%). This table evaluates **HoT**'s effectiveness on two LLMs (Qwen2.5:7b-instruct and DeepSeek-V3). Evaluations are conducted on GSM8K (mathematics) and OpenBookQA (logic). Results indicate **HoT** consistently improves reasoning accuracy for both LLMs, demonstrating its model-agnostic ability to enhance reasoning in both small and large parameter LLMs.

valuable for improving the reasoning capabilities of smaller LLMs, while the consistent performance boosts for DeepSeek-V3 demonstrate that strong models can also benefit from **HoT**'s structured resolution. These results highlight **HoT**'s general effectiveness across LLM's scales and task types.

### 4.2.3 ROBUSTNESS TESTING.

Robustness testing is performed using Total Variance (TV) and Instance Variance Mean (IVM) metrics on 300 runs with parameters M=300, N=10, B=100 and R=3, comparing the stability of **HoT** with CoT. We selected GSM8K as the dataset. CoT serves as the baseline with a Total Variance (TV) of 144.5 and an Instance Variance Mean (IVM) of 34.7. CoT-SC shows a higher TV of 155.0 (+7.27%) but a lower IVM of 22.7 (-34.58%). In contrast, **HoT** achieves a lower TV of 98.0 (-32.18%) and an IVM of 19.1 (-44.96%). Visual analysis in Figure3 further confirms that **HoT** exhibits substantially improved stability in inference results. TV quantifies global variability across multiple problem blocks. **HoT**'s lower TV indicates stronger robustness regardless of input distribution. IVM measures robustness by averaging variance in accuracy across repeated trials on the same problem blocks. The reduction in IVM highlights **HoT**'s ability to produce consistent results for identical problems, alleviating the unstable output that often troubles unstructured and unreliable reasoning methods. For comparisons of the remaining datasets, see the Appendix A.2.

### 4.2.4 ABLATION STUDY.

Ablation Studies have proved the effectiveness of each module. As shown in Table3, when **HoT** removes Module1 and does not explicitly mention the conditions in any of the subsequent treatments, there is the most significant drop in performance; when **HoT** removes Module2 and does not explore methods and paths, there is a slight drop in performance; when **HoT** removes Mod-

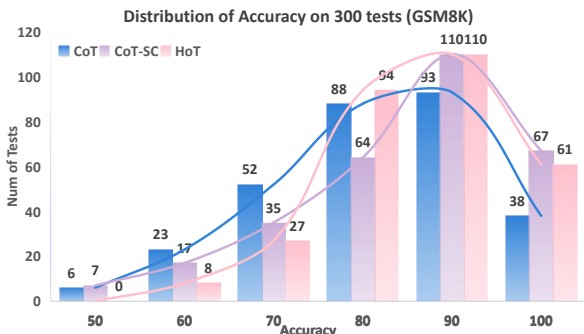

Figure 3: Robustness Comparison Between CoT, CoT-SC and **HoT** on the GSM8K Dataset. This figure presents the distribution of accuracy across 300 test runs for CoT, CoT-SC and **HoT** methods. The experiment was conducted on the Qwen2.5:7b-instruct model. It visually demonstrates that **HoT** exhibits significantly lower variance in inference results compared to CoT and CoT-SC, confirming **HoT**'s superior stability and consistency in repeated reasoning tasks.

| Methods | GSM8K | OpenBookQA |
|---|---|---|
| **HoT** (Full) | **86.3** | **91.5** |
| **HoT** (w/o Module1) | 82.2 | 89.5 |
| **HoT** (w/o Module2) | 85.5 | 89.7 |
| **HoT** (w/o Module3, 4) | 82.7 | 91.2 |

Table 3: Ablation Study on **HoT** Core Components (%). This table assesses the contribution of each **HoT** module by removing them individually. Results show removing Module 1 (Condition Extraction) causes the largest performance drop (4.1% on GSM8K, 2.0% on OpenBookQA), while removing other modules leads to smaller declines. This confirms that all modules are integral to **HoT**'s success, with condition extraction being critical for maintaining reasoning accuracy.

ule3, 4 and does not conduct problem decomposition and aggregation, there is also a slight drop in performance. Without condition extraction and maintenance, the model sometimes ignores important information when reasoning, leading to biased results. Without the exploration of methods and paths, the model has a more inadequate understanding of the problem and lacks an understanding of the big picture. Decomposing and aggregating the problem can reduce the complexity of the demand solution problem. Since subproblems often require only linear reasoning (without the need to consider multiple scenarios in parallel), this allows for different treatments in different situations without contaminating each other, further improving the robustness of the solution.

## 5 CONCLUSION

**HoT** represents a pivotal advance in enabling reliable, condition-aware reasoning in LLMs. It integrates explicit condition extraction, strategic planning, adaptive decomposition, and structured aggregation into a cohesive prompt-based framework, directly addressing core limitations of existing methods—including error propagation, poor interpretability, and weak condition satisfaction. Experiments validate **HoT**'s effectiveness: it boosts performance across diverse LLMs, and demonstrates greater robustness. Its structured reasoning traces enhance auditability, making it ideal for high-stakes fields like engineering and decision support.

**HoT**'s success underscores a key insight: advancing LLM reasoning requires reimagining the process, not just scaling models or data. Future work will extend **HoT** to dynamic/uncertain conditions, integrate external subproblem verifiers, and apply it to real-world engineering tasks. Automating decomposition heuristics (tailored to problem and condition structure) is another promising direction. By providing a blueprint for structured, condition-aware reasoning, **HoT** lays the groundwork for more trustworthy LLMs in rigorous domains.

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

# A APPENDIX

## A.1 EXAMPLE OF HoT REASONING PROCESS

### A.1.1 QUESTION

Suppose the polynomial

$$f(x) = a_n x^n + a_{n-1} x^{n-1} + \cdots + a_2 x^2 + a_1 x + a_0$$

has integer coefficients, and its roots are distinct integers.

Given that $a_n = 2$ and $a_0 = 66$, what is the least possible value of $|a_{n-1}|$?

### A.1.2 SOLUTION

Since $f(x)$ has integer coefficients, the Integer Root Theorem tells us that all integer roots of $f(x)$ must divide the constant term $66 = 2 \cdot 3 \cdot 11$. Thus, the possible integer roots of $f(x)$ are

$$\pm 1, \ \pm 2, \ \pm 3, \ \pm 6, \ \pm 11, \ \pm 22, \ \pm 33, \ \pm 66.$$

Moreover, since we know that all roots of $f(x)$ are integers, we know that all roots of $f(x)$ appear in the list above.

Now we apply Vieta's formulas. The product of the roots of $f(x)$ is $(-1)^n \cdot \frac{a_0}{a_n}$, which is 33 or $-33$. Also, the sum of the roots is $-\frac{a_{n-1}}{a_n} = -\frac{a_{n-1}}{2}$. Thus, in order to minimize $|a_{n-1}|$, we should make the absolute value of the sum of the roots as small as possible, working under the constraint that the product of the roots must be 33 or $-33$.

We now consider two cases.

Case 1 is that one of $33, -33$ is a root, in which case the only other possible roots are $\pm 1$. In this case, the absolute value of the sum of the roots is at least $32$.

The alternative, Case 2, is that one of $11, -11$ is a root and one of $3, -3$ is a root. Again, the only other possible roots are $\pm 1$, so the absolute value of the sum of the roots is at least $11 - 3 - 1 = 7$, which is better than the result of Case 1. If the absolute value of the sum of the roots is 7, then $|a_{n-1}| = 7|a_n| = 7 \cdot 2 = 14$.

Therefore, we have shown that $|a_{n-1}| \geq 14$, and we can check that equality is achieved by

$$f(x) = 2(x+11)(x-3)(x-1)$$
$$= 2x^3 + 14x^2 - 82x + 66,$$

which has integer coefficients and integer roots. So the least possible value of $|a_{n-1}|$ is $\boxed{14}$.

### A.1.3 IO (INPUT/OUTPUT) METHOD

According to Vieta's formulas, the product of the roots $r_1, r_2, \ldots, r_n$ is related to the coefficients by the formula $(-1)^n r_1 r_2 \cdots r_n = \frac{a_0}{a_n} = \frac{66}{2} = 33$. The coefficient $a_{n-1}$ is given by $a_{n-1} = -a_n \sum_{i=1}^n r_i = -2 \sum_{i=1}^n r_i$. To minimize $|a_{n-1}|$, we must minimize the absolute value of the sum of the roots. We need to find a set of distinct integer roots whose product is $\pm 33$. The integer factors of 33 are $\pm 1, \pm 3, \pm 11, \pm 33$.

To make the sum of the roots as close to zero as possible, we should select roots that cancel each other out. Consider the set of four distinct integer roots $\{1, -1, 3, -11\}$. The product of these roots is $(1)(-1)(3)(-11) = 33$. For this set of roots, the sum is $1 + (-1) + 3 + (-11) = -8$. This corresponds to a value of $a_{n-1} = -2 \times (-8) = 16$. Therefore, the least possible value of $|a_{n-1}|$ is $\boxed{16}$.

### A.1.4 HoT METHOD

CONDITION EXTRACTION

The first module of the process focuses on meticulously extracting and defining all constraints from the problem statement. The explicit constraints are that the polynomial $f(x)$ has **integer coefficients**, its roots are **distinct integers**, the leading coefficient is $a_n = 2$, and the constant term is $a_0 = 66$.

From these, several implicit constraints are derived. The relationship between coefficients and roots is governed by Vieta's formulas. This leads to two critical deductions:

- The product of the roots, adjusted for the leading coefficient, is fixed. Specifically, $(-1)^n \prod_{i=1}^n r_i = \frac{a_0}{a_n} = \frac{66}{2} = 33$. This means the product of the $n$ distinct integer roots must be either 33 (if $n$ is even) or $-33$ (if $n$ is odd).
- The coefficient $a_{n-1}$ is directly proportional to the sum of the roots: $a_{n-1} = -a_n \sum_{i=1}^n r_i = -2 \sum_{i=1}^n r_i$.

The ultimate goal is to find the minimum possible value of $|a_{n-1}|$, which translates to finding a set of distinct integer roots that satisfies the product constraint while making the absolute value of their sum as small as possible.

TREE EXPLORER

The second module explores various potential strategies for solving the problem. Three primary methods were identified and scored based on their perceived effectiveness and rigor:

- **Algebraic Approach (Score: 85):** This method, deemed the most promising, involves a systematic application of Vieta's formulas. The core idea is to enumerate all possible sets of distinct integer roots whose product is $\pm 33$, calculate the sum for each set, determine the corresponding $|a_{n-1}|$, and identify the minimum value. Its strength lies in its direct and exhaustive nature.

- **Combinatorial Optimization (Score: 75):** This approach frames the problem as minimizing the absolute sum of a set of integers under the constraint that their product is fixed. While conceptually sound, it was considered slightly less direct than the algebraic approach because it focuses more on the optimization aspect rather than a structured enumeration based on the factors of 33.
- **Analytical Approach (Score: 70):** This method proposed using calculus to treat $a_{n-1}$ as a function of the roots and find its minimum. This was rated the lowest because the problem's domain consists of discrete integers, making continuous calculus methods difficult and impractical to apply directly without significant adaptation.

The algebraic approach was selected as the most direct and reliable path to the solution.

ADAPTIVE DOMAIN DECOMPOSITION

The third module takes the chosen algebraic approach and breaks the problem down into a series of smaller, manageable subproblems or cases. The primary decomposition is based on the number of distinct integer roots, $n$. Since the product of the roots must be $\pm 33$, and $33 = 3 \times 11$, the number of roots is not predetermined but must be at least two. The analysis was therefore structured to check cases for different values of $n$.

The subproblems identified were:

- **Case n = 2 (Two roots):** The product of the roots, $r_1 r_2$, must be 33. All pairs of distinct integers with this product are examined.
- **Case n = 3 (Three roots):** The product of the roots, $r_1 r_2 r_3$, must be $-33$. All unique triplets of distinct integers with this product are investigated.
- **Case n = 4 (Four roots):** The product, $r_1 r_2 r_3 r_4$, must be 33. All quartets of distinct integers satisfying this are considered.
- **Cases n > 4:** These cases are also considered, but it's reasoned that adding more roots, especially pairs like $(1, -1)$, is the most efficient way to increase $n$ while controlling the sum.

SUBPROBLEM RESOLUTION AND AGGREGATION

The final module involves solving each subproblem defined in the previous stage and then aggregating the results to find the overall minimum value.

- **Solving for n=2:** The possible root sets whose product is 33 are $\{3, 11\}$ and $\{-3, -11\}$. Their sums are 14 and $-14$, respectively. This leads to $|a_{n-1}| = |-2 \times (\pm 14)| = 28$. Other pairs like $\{1, 33\}$ yield larger sums.
- **Solving for n=3:** The product of the roots must be $-33$. The system systematically explores combinations:
  - $\{-1, 3, 11\}$ has a sum of 13, giving $|a_{n-1}| = 26$.
  - $\{1, -3, 11\}$ has a sum of 9, giving $|a_{n-1}| = 18$.
  - $\{1, 3, -11\}$ has a sum of $-7$, giving $|a_{n-1}| = |-2 \times (-7)| = \mathbf{14}$.
  - $\{-1, -3, 11\}$ has a product of 33, not $-33$, so it's invalid for $n = 3$.
- **Solving for n=4:** The product must be 33. To keep the sum small, pairs of opposites are used. The set $\{1, -1, 3, -11\}$ has distinct roots, a product of 33, and a sum of $1 - 1 + 3 - 11 = -8$. This yields $|a_{n-1}| = |-2 \times (-8)| = 16$.
- **Aggregation and Final Answer:** The results from all valid cases are compared:
  - For n=2, the minimum $|a_{n-1}|$ is 28.
  - For n=3, the minimum $|a_{n-1}|$ is 14.
  - For n=4, the minimum $|a_{n-1}|$ is 16.

The overall minimum value found across all examined cases is 14. This occurs for the set of three distinct integer roots $\{1, 3, -11\}$. The reasoning correctly accounts for the parity of $n$ affecting the sign of the root product, a detail the other methods missed, leading them to incorrect conclusions.

**Final Answer:** $\boxed{14}$

---

**Algorithm 1 HoT** Reasoning Algorithm

---

**Input**: Initial question $Q$
**Output**: Final answer $A$
1: $\mathcal{C}_e, \mathcal{C}_i \leftarrow \text{ConditionExtraction}_{\text{LLM}}(Q)$
    //Extract explicit/implicit conditions
2: $\mathcal{P} \leftarrow \text{GeneratePaths}_{\text{LLM}}(Q, \mathcal{C}_e, \mathcal{C}_i)$
    //Generate solution paths
3: $\pi^* \leftarrow \underset{\pi_i \in \mathcal{P}}{\text{argmax}} \, \text{Score}_{\text{LLM}}(\pi_i)$
    //Select optimal path
4: $DecomposeFlag \leftarrow \text{NeedDecomposition}_{\text{LLM}}(\pi^*)$
5: **if** $DecomposeFlag$ == True **then**
6:    $\mathcal{Q}_{sub} \leftarrow \text{Decompose}_{\text{LLM}}(Q, \pi^*, \mathcal{C}_e, \mathcal{C}_i)$
    //Domain decomposition
7:    $\mathcal{A}_{sub} \leftarrow \varnothing$
8:    **for** $Q_i \in \mathcal{Q}_{sub}$ **do**
9:        $A_i \leftarrow \text{Solve}_{\text{LLM}}(Q_i, \pi^*, \mathcal{C}_e, \mathcal{C}_i)$
10:      $\mathcal{A}_{sub} \leftarrow \mathcal{A}_{sub} \cup \{A_i\}$
11:   **end for**
12:   $A \leftarrow \text{Aggregate}_{\text{LLM}}(\mathcal{A}_{sub}, Q, \mathcal{C}_e, \mathcal{C}_i)$
13: **else**
14:   $A \leftarrow \text{Solve}_{\text{LLM}}(Q, \pi^*, \mathcal{C}_e, \mathcal{C}_i)$
    //Direct resolution
15: **end if**
16: **return** $A$

---

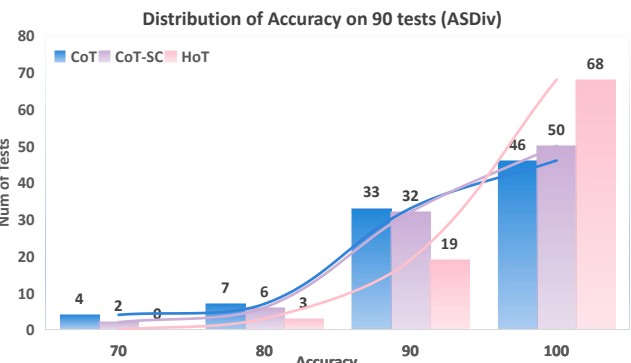

Figure 4: Robustness Comparison Between CoT, CoT-SC and **HoT** on the ASDiv Dataset. This figure presents the distribution of accuracy across 90 test runs for CoT, CoT-SC and **HoT** methods. The experiment was conducted on the Qwen2.5:7b-instruct model.

## A.2 COMPLETE ROBUSTNESS TESTING

### A.2.1 ROBUSTNESS TESTING ON ASDIV DATASET

Robustness testing on ASDiv Dataset is performed on 90 runs with parameters M=90, N=10, B=30 and R=3. CoT showes a TV of 64.8 and an IVM of 9.3. CoT-SC showes a TV of 51.4 (-20.68%) and an IVM of 7.9 (-15.05%). **HoT** achieves lower variance with a TV of 26.7 (-58.80%) and an IVM of 8.6 (-7.73%). Figure4 illustrates the trend.

### A.2.2 ROBUSTNESS TESTING ON SVAMP DATASET

Robustness testing on SVAMP Dataset is performed on 90 runs with parameters M=90, N=10, B=30 and R=3. CoT showes a TV of 117.1 and an IVM of 17.0. CoT-SC showes a TV of 82.7 (-29.38%) and an IVM of 14.8 (-12.94%). **HoT** achieves lower variance with a TV of 79.8 (-31.85%) and an IVM of 15.6 (-8.24%). Figure5 illustrates the trend.

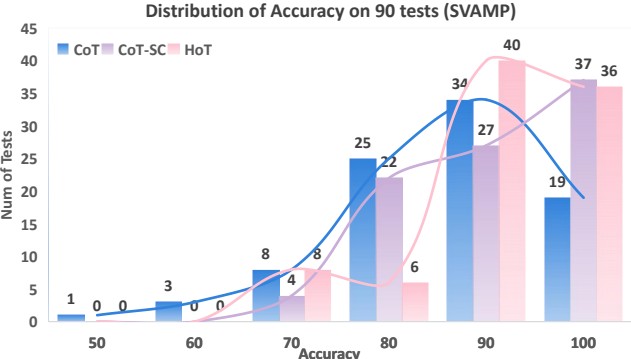

Figure 5: Robustness Comparison Between CoT, CoT-SC and **HoT** on the SVAMP Dataset. This figure presents the distribution of accuracy across 90 test runs for CoT, CoT-SC and **HoT** methods. The experiment was conducted on the Qwen2.5:7b-instruct model.

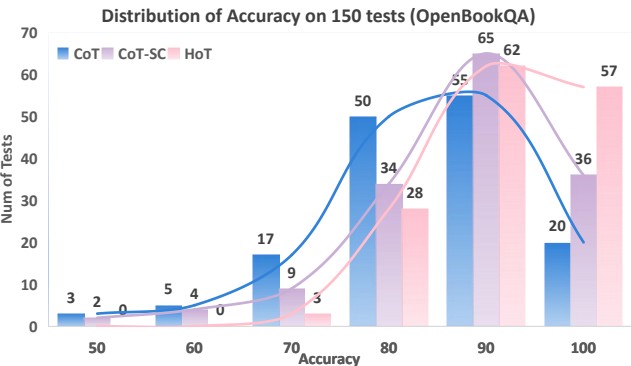

Figure 6: Robustness Comparison Between CoT, CoT-SC and **HoT** on the OpenBookQA Dataset. This figure presents the distribution of accuracy across 150 test runs for CoT, CoT-SC and **HoT** methods. The experiment was conducted on the Qwen2.5:7b-instruct model.

### A.2.3 ROBUSTNESS TESTING ON OPENBOOKQA DATASET

Robustness testing on OpenBookQA Dataset is performed on 150 runs with parameters M=150, N=10, B=50 and R=3. CoT showes a TV of 117.2 and an IVM of 16.0. CoT-SC showes a TV of 110.2 (-5.97%) and an IVM of 14.7 (-8.13%). **HoT** achieves lower variance with a TV of 62.3 (-46.84%) and an IVM of 10.2 (-36.25%). Figure6 illustrates the trend.

### A.2.4 ROBUSTNESS TESTING ON STRATEGYQA DATASET

Robustness testing on StrategyQA Dataset is performed on 300 runs with parameters M=300, N=10, B=100 and R=3. CoT showes a TV of 124.3 and an IVM of 32.2. CoT-SC showes a TV of 103.8 (-16.49%) and an IVM of 20.7 (-35.71%). **HoT** achieves lower variance with a TV of 87.2 (-29.85%) and an IVM of 18.2 (-43.48%). Figure7 illustrates the trend.

### A.3 COST ANALYSIS

This section quantifies the trade-off between performance and efficiency of **HoT** and its variant, compared to baseline reasoning methods. The cost analysis utilized the results from Table1 and the corresponding average resource consumption values calculated for each individual question. To address large disparities in token consumption across methods, the token count is visualized on a log10 scale. This ensures clear separation of data points

Figure8 presents the relationship between accuracy and computational resource consumption for **HoT**, **HoT \***, and six baseline methods (CoT, Random-CoT, CoT-SC, ReAct, IAP, AoT). **HoT**

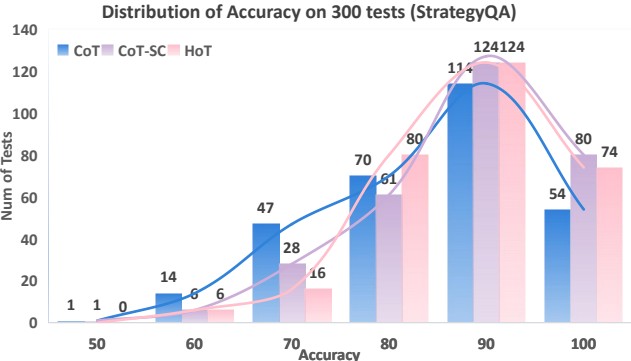

Figure 7: Robustness Comparison Between CoT, CoT-SC and **HoT** on the StrategyQA Dataset. This figure presents the distribution of accuracy across 300 test runs for CoT, CoT-SC and **HoT** methods. The experiment was conducted on the Qwen2.5:7b-instruct model.

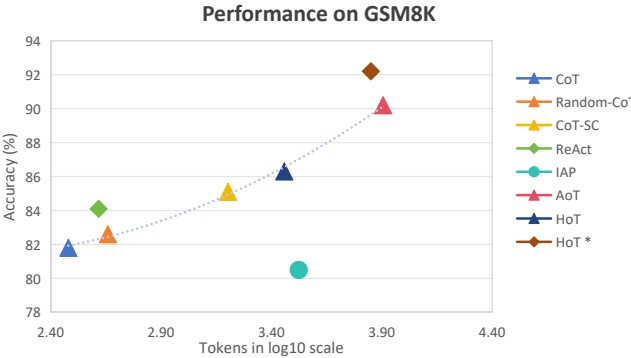

Figure 8: Accuracy vs. Computational Resource Consumption (Total Token Count in log10 Scale) Across Reasoning Methods

strikes a balance between accuracy and efficiency, aligning with the growth trend observed in most baseline methods. Meanwhile, **HoT \*** demonstrates excellent performance by significantly boosting accuracy while increasing computational resource consumption.

### A.4 PROMPTS OF **HoT** IN MATHEMATICS

In this section, we introduce prompts for each module of **HoT** in Mathematics (GSM8K).

#### A.4.1 CONDITION EXTRACTION

```
async def _condition_extraction(self, question: str) -> Dict[str, Any]:
    prompt = f"""
        You are a world-class mathematician and mathematical logician.
        You are intelligent, rigorous, and cautious.
        You always reason step by step, consider all relevant conditions.
        You think in terms of structure, symmetry, and mathematical
            principles, and never skip important logical steps.
        You aim to find a complete and correct solution, not just an
            answer.
        You THINK CLEARLY, STRUCTURALLY, AND DEEPLY.
        Analyze this math problem and extract ALL conditions:
        problem:{question}
        Notice:
        1. Identify explicit conditions (directly stated in the problem)
        2. Derive implicit conditions (e.g., denominators > 0, square
            roots > 0, log arguments > 0)
```

```
14          3. Determine domain restrictions based on mathematical principles
15          4. Identify range limitations from problem context
16          5. Extract physical meaning conditions (e.g., length > 0,
               probability in [0,1])
17          Output JSON format:
18          {{
19              "explicit": ["condition1", "condition2"],
20              "implicit": ["condition1", "condition2"],
21              "notes": "Additional analysis notes"
22          }}
23      """
24      for attempt in range(self.config.max_retries):
25          try:
26              response = await self.llm.generate(prompt, response_format="
                  json_object")
27              data = json.loads(response)
28              if not isinstance(data, dict):
29                  continue
30              conditions = {
31                  "explicit": data.get("explicit", []),
32                  "implicit": data.get("implicit", []),
33                  "notes": data.get("notes", "")
34              }
35              if not (conditions["explicit"] or conditions["implicit"]):
36                  continue
37              return conditions
38          except (json.JSONDecodeError, AttributeError) as e:
39              continue
40      return {
41          "explicit": ["Default explicit condition"],
42          "implicit": ["Default implicit condition"],
43          "notes": "Fallback conditions"
44      }
```

Listing 1: Condition Extraction Prompt in Mathematics

### A.4.2 TREE EXPLORER

```
1  async def _tree_explorer(self, question: str) -> List[Dict[str, Any]]:
2      prompt = f"""
3          You are a world-class mathematician and mathematical logician.
4          You are intelligent, rigorous, and cautious.
5          You always reason step by step, consider all relevant conditions.
6          You think in terms of structure, symmetry, and mathematical
               principles, and never skip important logical steps.
7          You aim to find a complete and correct solution, not just an
               answer.
8          You THINK CLEARLY, STRUCTURALLY, AND DEEPLY.
9          Generate 3 distinct solution methods for:
10         {question}
11         Notice:
12         1. Employ different theoretical frameworks (algebraic, geometric,
               analytical, etc.)
13         2. Approach from fundamentally different perspectives
14         3. Vary implementation techniques significantly
15         4. Consider both conventional and innovative methods
16         5. Steps can be retained as ideas only, without exact
               calculations
17         6. Pay attention to the mathematical expressions in the questions
               and understand them correctly
18         7. examine carefully the subject matter
19         For each method, provide:
20         - Clear description of the mathematical approach
21         - Step-by-step implementation plan
```

```
22          - Effectiveness score (0-100) based on:
23            * Mathematical rigor
24            * Computational feasibility
25            * Logical completeness
26            * Efficiency
27
28          Output JSON format:
29          {{
30              "methods": [
31                  {{
32                      "description": "Method description",
33                      "steps": ["step1", "step2"],
34                      "score": 0-100,
35                      "score_reason": "Scoring justification"
36                  }}
37              ]
38          }}
39      """
40
41      for attempt in range(self.config.max_retries):
42          try:
43              response = await self.llm.generate(prompt, response_format="
                  json_object")
44              response = response.strip()
45              data = json.loads(response)
46              if not isinstance(data, dict) or "methods" not in data:
47                  raise ValueError("Invalid structure: missing 'methods'
                      key")
48              methods = data["methods"]
49              if len(methods) != 3:
50                  raise ValueError(f"Expected 3 methods, got {len(methods)}
                      ")
51              required_keys = {"description", "steps", "score", "
                  score_reason"}
52              for method in methods:
53                  if not all(k in method for k in required_keys):
54                      raise ValueError("Missing required keys in method")
55                  if not isinstance(method["steps"], list):
56                      raise ValueError("Steps must be a list")
57              return sorted(methods, key=lambda x: -x["score"])
58          except (json.JSONDecodeError, ValueError, KeyError) as e:
59              if attempt == self.config.max_retries - 1:
60                  return []
61              continue
62      return []
```

Listing 2: Tree Explorer Prompt in Mathematics

### A.4.3 ADAPTIVE DOMAIN DECOMPOSITION

```
1  async def _adaptive_domain_decomposition(self, method: str, steps: List[
       str]) -> Dict[str, Any]:
2      prompt = f"""
3          You are a world-class mathematician and mathematical logician.
4          You are intelligent, rigorous, and cautious.
5          You always reason step by step, consider all relevant conditions.
6          You think in terms of structure, symmetry, and mathematical
               principles, and never skip important logical steps.
7          You aim to find a complete and correct solution, not just an
               answer.
8          You THINK CLEARLY, STRUCTURALLY, AND DEEPLY.
9          Determine if this solution requires classification:
10         Method: {method}
11         Steps: {steps}
```

```
12          Notice:
13          1. Identify parameter dependencies requiring discussion
14          2. Detect interval-specific elements (absolute values, piecewise
                 functions)
15          3. Recognize domain-specific computation requirements
16          4. Flag multiple solution sets needing verification
17          5. Pay attention to the mathematical expressions in the questions
                 and understand them correctly
18          6. examine carefully the subject matter
19          If classification needed, provide:
20          - Comprehensive case descriptions
21          - Precise mathematical conditions for each case
22          - Clear boundary conditions
23          Output JSON format:
24          {{
25              "need_classify": true/false,
26              "reason": "Classification rationale",
27              "cases": [
28                  {{
29                      "description": "Case description",
30                      "conditions": {{"parameter": "value_range"}}
31                  }}
32              ]
33          }}
34      """
35      response = await self.llm.generate(prompt, response_format="
            json_object")
36      try:
37          data = json.loads(response)
38          return {
39              "need_classify": data.get("need_classify", False),
40              "reason": data.get("reason", ""),
41              "cases": data.get("cases", [])
42          }
43      except json.JSONDecodeError:
44          return {"need_classify": False, "reason": "Parse failed", "cases"
                : []}
```

Listing 3: Adaptive Domain Decomposition Prompt in Mathematics

### A.4.4 SUBQUESTION RESOLUTION AND AGGREGATION

```
1  async def _resolution(self, node_id: str) -> Optional[Dict[str, Any]]:
2      node = self.nodes[node_id]
3      root_node = self.nodes[node.path[0]] if node.path else node
4      original_question = root_node.method.get("description", "Original
           problem")
5
6      prompt = f"""
7          You are a world-class mathematician and mathematical logician.
8          You are intelligent, rigorous, and cautious.
9          You always reason step by step, consider all relevant conditions.
10         You think in terms of structure, symmetry, and mathematical
               principles, and never skip important logical steps.
11         You aim to find a complete and correct solution, not just an
               answer.
12         You THINK CLEARLY, STRUCTURALLY, AND DEEPLY.
13         You are a meticulous mathematical problem solver executing this
               solution:
14
15         Original Problem: {original_question}
16         Steps: {node.steps}
17         conditions: {node.conditions}
18
```

```
19          As an executor, you must:
20          - Explicitly verify all conditions
21          - Show complete mathematical reasoning
22          - Include standalone line: "Final Answer: \\boxed{{answer}}"
23          - Ensure your answer directly responds to the question asked
24          - The final answer should be one exact number
25          - Not all conditions can serve as the conditions for solving
                problems. We should answer according to the problems
26      """
27      response = await self.llm.generate(prompt)
28      answer = self._extract_answer(response)
29      if answer:
30          node.answer = answer
31          node.state = "solved"
32          return {
33              "node_id": node_id,
34              "response": response,
35              "answer": answer
36          }
37      return None
38
39  async def _aggregation(self, solutions: List[Dict[str, Any]]) -> str:
40      if not solutions:
41          return "No valid solutions found"
42      original_question = None
43      for sol in solutions:
44          node = self.nodes[sol["node_id"]]
45          if hasattr(node, 'original_question'):
46              original_question = node.original_question
47              break
48      if original_question is None:
49          first_node = self.nodes[solutions[0]["node_id"]]
50          path = first_node.path
51          if path:
52              root_node_id = path[0]
53              root_node = self.nodes.get(root_node_id)
54              if root_node:
55                  original_question = root_node.method.get("description", "
                      Original problem")
56      if original_question is None:
57          original_question = "Original problem (reconstructed from context
              )"
58          if solutions[0]["response"]:
59              match = re.search(r'Original Problem[:\s]*(.+?)\nSteps:',
                  solutions[0]["response"])
60              if match:
61                  original_question = match.group(1).strip()
62      if len(solutions) == 1:
63          return solutions[0]["answer"]
64      unique_answers = {sol["answer"] for sol in solutions}
65      if len(unique_answers) == 1:
66          return solutions[0]["answer"]
67      solutions_text = "\n\n".join(
68          f"Solution {i+1} (Node: {sol['node_id']}):\n"
69          f"Answer: {sol['answer']}\n"
70          f"Approach: {self.nodes[sol['node_id']].method['description']}\n"
71          f"conditions: {self.nodes[sol['node_id']].conditions}\n"
72          f"Reasoning Excerpt:\n{sol['response'][:300]}...\n"
73          for i, sol in enumerate(solutions)
74      )
75      prompt = f"""
76          You are a world-class mathematician and mathematical logician.
77          You are intelligent, rigorous, and cautious.
78          You always reason step by step, consider all relevant conditions.
```

```
79          You think in terms of structure, symmetry, and mathematical
                principles, and never skip important logical steps.
80          You aim to find a complete and correct solution, not just an
                answer.
81          You THINK CLEARLY, STRUCTURALLY, AND DEEPLY.
82          Synthesize these solutions for the original problem:
83          Original Problem: {original_question}
84          Proposed Solutions:
85          {solutions_text}
86          As an analyst, you must:
87          1. FIRST verify which solution(s) correctly answer the original
                question
88          2. Compare mathematical consistency with the original problem
                statement
89          3. Evaluate which approach best satisfies all conditions
90          4. Combine elements from multiple solutions ONLY if
                mathematically valid
91          5. Provide clear justification for your selection
92          6. Mark final answer with \\boxed{{}}
93          7. Include standalone line: "Aggregated Answer: answer"
94          Critical Analysis Guidelines:
95          - The solution MUST directly answer the original question as
                stated
96          - Prioritize mathematical correctness over elegance
97          - Reject solutions that violate any explicit conditions
98          - Verify all intermediate calculations are sound
99          - Ensure the final answer format matches what the problem
                requires
100         """
101         response = await self.llm.generate(prompt)
102         return self._extract_answer(response) or "Aggregation failed"
```

Listing 4: Subquestion Resolution and Aggregation Prompt in Mathematics

## A.5   PROMPTS OF **HoT** IN LOGIC

In this section, we introduce prompts for each module of **HoT** in Logic (OpenBookQA).

### A.5.1   CONDITION EXTRACTION

```
1  async def _condition_extraction(self, question: str, options: Dict[str,
       str]) -> Dict[str, Any]:
2      prompt = f"""
3          You are a top expert in formal logic, critical thinking, and
                argument analysis.
4          You are precise, rational, and skeptical.
5          You always examine each statement carefully, identify premises
                and conclusions, and evaluate logical validity step by step.
6          You avoid unwarranted assumptions, think in terms of logical
                consequences, and eliminate invalid options with sound
                reasoning.
7          You aim to reach conclusions based only on evidence and logic.
8          You THINK SLOWLY, CAREFULLY, AND LOGICALLY.
9          Analyze this question and extract key conditions:
10
11         Question: {question}
12         Options:
13         A. {options['A']}
14         B. {options['B']}
15         C. {options['C']}
16         D. {options['D']}
17
18         Identify:
19         1. Explicit conditions (directly stated)
```

```
20        2. Implicit conditions (logical implications)
21        3. Key terms and their relationships
22        4. Spatial/temporal relationships if present
23        5. Any conditional statements
24
25        Output JSON format:
26        {{
27            "explicit": ["list", "of", "conditions"],
28            "implicit": ["list", "of", "conditions"],
29            "key_terms": ["term1", "term2"],
30            "notes": "Analysis summary"
31        }}
32    """
33    for attempt in range(self.config.max_retries):
34        try:
35            response = await self.llm.generate(prompt, response_format="
                json_object")
36            return json.loads(response)
37        except:
38            continue
39    return {
40        "explicit": [],
41        "implicit": [],
42        "key_terms": [],
43        "notes": "Failed to extract conditions"
44    }
```

Listing 5: Condition Extraction Prompt in Logic

### A.5.2 TREE EXPLORER

```
1  async def _tree_explorer(self, question: str, options: Dict[str, str]) ->
       List[Dict]:
2      options_text = "\n".join([f"{k}. {v}" for k, v in options.items()])
3      prompt = f"""
4          You are a top expert in formal logic, critical thinking, and
               argument analysis.
5          You are precise, rational, and skeptical.
6          You always examine each statement carefully, identify premises
               and conclusions, and evaluate logical validity step by step.
7          You avoid unwarranted assumptions, think in terms of logical
               consequences, and eliminate invalid options with sound
               reasoning.
8          You aim to reach conclusions based only on evidence and logic.
9          You THINK SLOWLY, CAREFULLY, AND LOGICALLY.
10         Generate 3 distinct solution approaches for this question:
11         Question: {question}
12         Options:
13         {options_text}
14         For each approach, provide:
15         - Clear description of the reasoning strategy
16         - Key steps to implement the approach
17         - Confidence score (0-100) based on:
18         * Logical soundness
19         * Coverage of options
20         * Appropriate use of deductive/inductive reasoning
21         * Clarity of reasoning steps
22         Output JSON format:
23         {{
24             "methods": [
25                 {{
26                     "description": "Approach description",
27                     "steps": ["step1", "step2"],
28                     "score": 0-100,
```

```
29                       "score_reason": "Scoring justification"
30                   }}
31               ]
32           }}
33       """
34       for attempt in range(self.config.max_retries):
35           try:
36               response = await self.llm.generate(prompt, response_format="
                   json_object")
37               response = response.strip()
38               if response.startswith("```json"):
39                   response = response[7:-3].strip()
40               elif response.startswith("```"):
41                   response = response[3:-3].strip()
42               data = json.loads(response)
43               if not isinstance(data, dict) or "methods" not in data:
44                   raise ValueError("Invalid structure: missing 'methods'
                       key")
45               methods = data["methods"]
46               if len(methods) < 2:
47                   raise ValueError(f"Expected at least 2 methods, got {len(
                       methods)}")
48               required_keys = {"description", "steps", "score", "
                   score_reason"}
49               for method in methods:
50                   if not all(k in method for k in required_keys):
51                       raise ValueError("Missing required keys in method")
52                   if not isinstance(method["steps"], list):
53                       raise ValueError("Steps must be a list")
54               return sorted(methods, key=lambda x: -x["score"])
55           except (json.JSONDecodeError, ValueError, KeyError) as e:
56               if attempt == self.config.max_retries - 1:
57                   return []
58               continue
59       return []
```

Listing 6: Tree Explorer Prompt in Logic

### A.5.3 ADAPTIVE DOMAIN DECOMPOSITION

```
1  async def _adaptive_domain_decomposition(self, method: str, question: str
       , options: Dict[str, str]) -> Dict[str, Any]:
2      options_text = "\n".join([f"{k}. {v}" for k, v in options.items()])
3      prompt = f"""
4          You are a top expert in formal logic, critical thinking, and
               argument analysis.
5          You are precise, rational, and skeptical.
6          You always examine each statement carefully, identify premises
               and conclusions, and evaluate logical validity step by step.
7          You avoid unwarranted assumptions, think in terms of logical
               consequences, and eliminate invalid options with sound
               reasoning.
8          You aim to reach conclusions based only on evidence and logic.
9          You THINK SLOWLY, CAREFULLY, AND LOGICALLY.
10         Determine if this solution approach requires case classification:
11         Solution Approach: {method}
12         Question: {question}
13         Options:
14         {options_text}
15         Consider:
16         1. Does the question contain multiple scenarios or cases?
17         2. Are there conditional statements that create distinct
               possibilities?
18         3. Do the options represent different logical paths?
```

```
19          4. Would different initial assumptions lead to different
                solutions?
20          If classification needed, provide:
21          - Comprehensive case descriptions
22          - Precise conditions for each case
23          - Expected outcomes
24          Output JSON format:
25          {{
26              "need_classify": true/false,
27              "reason": "Classification rationale",
28              "cases": [
29                  {{
30                      "description": "Case description",
31                      "conditions": {{"parameter": "value_range"}}
32                  }}
33              ]
34          }}
35      """
36      try:
37          response = await self.llm.generate(prompt, response_format="
                json_object")
38          data = json.loads(response)
39          return data
40      except:
41          return {
42              "need_classify": False,
43              "reason": "Analysis failed",
44              "cases": []
45          }
```

Listing 7: Adaptive Domain Decomposition Prompt in Logic

### A.5.4 SUBQUESTION RESOLUTION AND AGGREGATION

```
1  async def _resolution(self, node_id: str) -> Optional[Dict[str, Any]]:
2      node = self.nodes[node_id]
3      context = f"Question: {node.question}\nOptions:\n"
4      for opt, text in node.options.items():
5          context += f"{opt}. {text}\n"
6      context += f"\nSolution Approach: {node.method['description']}\n"
7      context += f"conditions: {json.dumps(node.conditions, indent=2)}\n"
8      prompt = f"""
9          You are a top expert in formal logic, critical thinking, and
                argument analysis.
10          You are precise, rational, and skeptical.
11          You always examine each statement carefully, identify premises
                and conclusions, and evaluate logical validity step by step.
12          You avoid unwarranted assumptions, think in terms of logical
                consequences, and eliminate invalid options with sound
                reasoning.
13          You aim to reach conclusions based only on evidence and logic.
14          You THINK SLOWLY, CAREFULLY, AND LOGICALLY.
15          Solve this question using the specified approach:
16          {context}
17          Reasoning Steps:
18          1. Strictly follow the provided approach: {node.method['
                description']}
19          2. Execute each step: {', '.join(node.method['steps'])}
20          3. Consider all conditions
21          4. Evaluate each option systematically
22          5. Provide clear justification for inclusion/exclusion
23          6. Select the best answer
24          Output Requirements:
25          - End your response with: "Final Answer: [OPTION]"
```

```
26          - Use \boxed{{[OPTION]}} to denote your answer
27          - Your answer must be A, B, C, or D
28      """
29      response = await self.llm.generate(prompt)
30      answer = self._extract_answer(response)
31      if answer:
32          node.answer = answer
33          node.state = "solved"
34          return {
35              "node_id": node_id,
36              "response": response,
37              "answer": answer
38          }
39      return None
40
41  async def _aggregation(self, solutions: List[Dict[str, Any]]) -> str:
42      if not solutions:
43          return "X"  # Invalid answer
44      if len(solutions) == 1:
45          return solutions[0]["answer"]
46      answers = [s["answer"] for s in solutions]
47      if len(set(answers)) == 1:
48          return answers[0]
49      solutions_text = ""
50      for i, sol in enumerate(solutions):
51          node = self.nodes[sol["node_id"]]
52          solutions_text += f"\n\nSolution {i+1} (Node {sol['node_id']}):"
53          solutions_text += f"\nApproach: {node.method['description']}"
54          solutions_text += f"\nconditions: {json.dumps(node.conditions,
                  indent=2)}"
55          solutions_text += f"\nAnswer: {sol['answer']}"
56          solutions_text += f"\nReasoning Excerpt:\n{sol['response'][:]}...
                  "
57
58      prompt = f"""
59          You are a top expert in formal logic, critical thinking, and
              argument analysis.
60          You are precise, rational, and skeptical.
61          You always examine each statement carefully, identify premises
              and conclusions, and evaluate logical validity step by step.
62          You avoid unwarranted assumptions, think in terms of logical
              consequences, and eliminate invalid options with sound
              reasoning.
63          You aim to reach conclusions based only on evidence and logic.
64          You THINK SLOWLY, CAREFULLY, AND LOGICALLY.
65          Synthesize these approaches:
66
67          {solutions_text}
68
69          Instructions:
70          1. Analyze all solutions and their approaches
71          2. Identify the most reliable reasoning
72          3. Verify consistency with conditions
73          4. Select the best overall answer
74          5. Output format: \boxed{{[ANSWER]}}
75      """
76      response = await self.llm.generate(prompt)
77      return self._extract_answer(response) or "X"
```

Listing 8: Subquestion Resolution and Aggregation Prompt in Logic

## A.6 IMPLEMENTATION DETAILS

### A.6.1 REACT

We adopt a *few-shot ReAct-style* prompting strategy inspired by the ReAct framework. While the original ReAct framework combines language model reasoning with external tool-use, our implementation simplifies this structure by eliminating actual API calls or tool integration. Instead, we simulate both reasoning (`Thought`) and acting (`Action`) steps purely within natural language, forming a lightweight and deployable version suitable for standard API-based model access.

**Comparison with Original ReAct** The original ReAct framework relies on dynamic tool use: the model emits an `Action`, receives an `Observation`, and continues reasoning based on this feedback loop. This mechanism enhances performance on tasks requiring retrieval or real-time computation.

In contrast, our ReAct-style prompting:

- Requires no tool infrastructure or external observation integration.
- Can be used directly with closed-source APIs, enabling plug-and-play reasoning for both mathematical and commonsense questions.
- Emphasizes interpretability through explicit intermediate reasoning chains and simulated actions.

Thus, our method trades the dynamism of tool interaction for broad compatibility and simplicity, enabling structured reasoning with minimal implementation overhead. This makes it well-suited as a practical baseline for both mathematical and logical benchmarks.

**Original Format:**

> **Question:** The natural language query to be solved.
> **Thought:** Intermediate steps expressed as natural language reasoning.
> **Action (optional):** If needed, simulate tool-use as internal calculation.
> **Final Answer:** Explicit, boxed or stated final response.

**Example 1 (GSM8K-style Arithmetic Reasoning):**

> **Question:** Farmer Brown has 20 animals on his farm, all either chickens or cows. They have a total of 70 legs altogether. How many of the animals are chickens?
> **Thought:** Let $C$ be the number of chickens. Then the number of cows is $20 - C$. Chickens have 2 legs, cows have 4. So total legs $= 2C + 4(20 - C) = 70$.
> **Thought:** Simplify the equation: $2C + 80 - 4C = 70 \Rightarrow -2C = -10 \Rightarrow C = 5$.
> **Final Answer:** $\boxed{5}$

**Example 2 (OpenBookQA-style Commonsense Reasoning)**

> **Question:** Some animals use a liquid coming from their skin to adjust to
> **Choices:** A) cold    B) water    C) heat    D) humidity
> **Thought:** Many animals sweat to regulate their body temperature when it is hot. Sweat is a liquid that comes from the skin.
> **Final Answer:** $\boxed{C}$

### A.6.2 INSTANCE-ADAPTIVE PROMPTING (IAP)

We implement the IAP strategy with two variants: Majority Vote (IAP-mv) and Sequential Substitution (IAP-ss). The implementation builds upon the Qwen2.5:7b-instruct language model accessed via the Ollama API. Following the methodology, we employ nine distinct prompt templates:

```
candidates = [
    """Let's think step by step."""
```

```
3      """First,"""
4      """The answer is after the proof."""
5      """Before we dive into the answer,"""
6      """Let's solve this problem by splitting it into steps."""
7      """Let's think about this logically."""
8      """It's a beautiful day."""
9      """Don't think. Just feel."""
10     """By the fact that the earth is round,"""
11 ]
```

Listing 9: IAP Prompt Candidates

For each prompt-question pair, we compute a composite saliency score:

$$S = \lambda_1 I_{qp} + \lambda_2 I_{qr} + \lambda_3 I_{pr} \tag{10}$$

where:

- $I_{qp}$: Question-to-prompt information flow
- $I_{qr}$: Question-to-rationale information flow
- $I_{pr}$: Prompt-to-rationale information flow
- $\lambda_1 = 0.4$, $\lambda_2 = 0.4$, $\lambda_3 = 0.2$: Weighting parameters (summing to 1)

### A.6.3 MAJORITY VOTE (IAP-MV)

1. Generate responses using all nine prompts
2. Select top-3 responses by composite saliency score $S$
3. Apply majority voting on the extracted answers
4. Return the most frequent answer among top responses

### A.6.4 SEQUENTIAL SUBSTITUTION (IAP-SS)

1. Iterate through prompts in predefined order
2. For each prompt:
   (a) Generate response and compute $S$
   (b) If $S \geq \theta$ (threshold = $5.5 \times 10^{-6}$), return answer
3. If no prompt meets threshold, return last generated answer

### A.7 USE OF LARGE LANGUAGE MODELS

In accordance with the ICLR 2026 policy on the disclosure of Large Language Models (LLMs), we detail our use of LLMs in the preparation of this paper. LLMs were employed in a supportive capacity only, and their contributions were limited to non-core aspects of the work. All scientific content, including the conceptualization of the Holon-of-Thought (**HoT**) framework, methodology design, experimental setup, data analysis, and conclusions, was entirely developed by the human authors.

We used LLMs (specifically GPT-4) to aid in polishing the writing of the manuscript. LLMs were prompted to suggest improvements to sentence structure, clarity, and grammatical accuracy in drafts of sections such as the abstract, introduction, related work, and methodology. For example, we provided raw paragraphs and asked the model to rephrase for conciseness while preserving the original meaning.

LLMs were also used for retrieval and discovery purposes to identify potential related methods and literature. LLMs assisted in generating lists of potential citations by summarizing search queries related to "robustness in LLM reasoning" or "prompt-based decomposition techniques." This accelerated the discovery process, allowing us to quickly identify key works.

