# OpenReview forum: "Holon-of-Thought: Improving robustness in Large Language Models via structured framework"
_ICLR.cc/2026/Conference — ICLR 2026 Conference Withdrawn Submission_

### Official Review · Reviewer_cRbi · 2025-10-18

**Soundness:** 3
**Presentation:** 3
**Contribution:** 2
**Rating:** 2
**Confidence:** 4

**Summary:**

This paper introduces Holon-of-Thought (HoT), a framework designed to enhance the robustness of large language models (LLMs). The HoT framework comprises the following modules: Condition Extraction, Tree Explorer, Path Scoring, Optimal Path Selection, Adaptive Domain Decomposition, and Resolution and Aggregation. Experiments on the math and logic dataset demonstrate that HoT improves upon a set of baselines.

**Strengths:**

The paper is clearly written and well-structured, making it easy to follow.

**Weaknesses:**

- Motivation: The motivation for this work raises some concerns. Numerous studies have already addressed the rigor of LLM reasoning by incorporating external symbolic solvers, which provide strong guarantees of logical soundness and rigorousness. The authors criticize LLMs’ limitations in rigorous reasoning and conditional following, yet their proposed framework relies entirely on LLMs with prompt engineering, an approach that still lacks formal guarantees of rigor.

- Methodological Justification: The rationale behind the framework’s design choices is not well explained. Why were these specific modules or processes selected? What underlying principles or hypotheses motivated the design? At present, the methodology reads more like a technical implementation report, without providing conceptual insights into why or how it achieves improvement.

- Empirical Performance: The reported performance gains on most benchmarks appear minor.

- Analysis Depth: The paper would benefit from deeper qualitative or diagnostic analysis to help readers better understand the framework’s behavior, strengths, and limitations.

**Questions:**

- What is the motivation of HoT*?

---

### Official Review · Reviewer_b3J1 · 2025-10-29

**Soundness:** 2
**Presentation:** 2
**Contribution:** 3
**Rating:** 4
**Confidence:** 4

**Summary:**

The structured reasoning framework "Holon-of-Thought (HoT)" proposed in this paper addresses core pain points of Large Language Models (LLMs) in rigorous reasoning tasks, such as error propagation, hallucinatory outputs, and oversimplified handling of complex problems. It designs a four-stage pipeline of "Condition Extraction - Path Exploration - Adaptive Decomposition - Aggregation". The experimental part covers two typical types of reasoning tasks: mathematical tasks (GSM8K, ASDiv, SVAMP) and logical tasks (OpenBookQA, StrategyQA).

**Strengths:**

1. The innovation is clear: the paper profoundly analyzes the root causes of reasoning errors in Large Language Models (LLMs) (relying on statistical correlations rather than formal logic during training, and neglecting explicit/implicit conditions), and proposes a core design of "conditions throughout the entire process + sub-problem isolation" in a targeted manner.
2. The experimental design is rigorous: the experimental setup covers multiple task types (mathematics + logic), multiple models (Qwen2.5:7b-instruct, DeepSeek-V3), and multiple baselines (CoT, CoT-SC, ReAct, etc.). It adopts a two-dimensional evaluation of "accuracy + robustness (TV/IVM)" and verifies the necessity of the four core modules through ablation experiments.
3. The engineering practicality is strong: HoT is designed as a "model-agnostic" framework, which is implemented only through prompt engineering or lightweight API calls, without the need for retraining LLMs, thus avoiding the high computing power cost of traditional reasoning enhancement methods (such as Fine-tuning).

**Weaknesses:**

1. The "binary classification decision logic" (DecomposeFlag=True/False) of the Adaptive Decomposition module does not provide specific decision-making criteria.
2. In the Tree Explorer module, the "path evaluation criteria" only mention "anticipated accuracy, operational feasibility, and computational complexity," lacking a specific definition of Score_{LLM}.
3. There is confusion regarding performance: it seems that the HoT method itself cannot bring significant performance improvement, and the performance improvement mainly comes from the voting mechanism? If so, would other methods combined with the voting mechanism perform better?

**Questions:**

None.

---

### Official Review · Reviewer_dvti · 2025-10-30

**Soundness:** 2
**Presentation:** 3
**Contribution:** 1
**Rating:** 4
**Confidence:** 4

**Summary:**

The paper introduces Holon-of-Thought (HoT), a structured reasoning framework for Large Language Models (LLMs). It aims to improve robustness, logical consistency, and condition adherence in reasoning tasks.
HoT consists of a four-stage pipeline:
1. Condition Extraction – explicitly identifies both explicit and implicit problem constraints.
2. Tree Explorer – generates and scores multiple reasoning paths, then selects the optimal one.
3. Adaptive Domain Decomposition – decides whether to decompose complex problems into subproblems.
4. Resolution and Aggregation – solves each subproblem independently and aggregates the results.

The paper benchmarks HoT against several reasoning frameworks (CoT, CoT-SC, ReAct, IAP, AoT) across five datasets (GSM8K, ASDiv, SVAMP, OpenBookQA, StrategyQA) and shows improvements in accuracy and stability. The authors further provide variance-based robustness metrics (TV, IVM) and ablation studies, demonstrating each module’s contribution.

**Strengths:**

1. HoT presents a well-structured and modular pipeline that highlights condition awareness and adaptive decomposition. The focus on robustness and logical consistency is both sensible and well-motivated.
2. The introduction of TV and IVM as robustness metrics is a valuable contribution, providing concrete measures for evaluating the stability of reasoning.
3. The experiments span diverse reasoning benchmarks and include comprehensive comparisons with multiple baselines. The ablation study offers clear insights into the contribution of each component, with Condition Extraction identified as the most influential.
4. The framework naturally improves interpretability by explicitly revealing condition sets and reasoning trajectories, allowing for clearer understanding and analysis of the model’s decision process.

**Weaknesses:**

1. While the framework is well-structured, the paper lacks a deeper theoretical justification or analysis of why the modular decomposition guarantees robustness. There is no formal reasoning or error-bound discussion—results are largely empirical. Meanwhile, the novelty of the article is relatively low. Trying different XoT methods has long been proven to be effective, but it does not bring about an essential improvement.
2. The selected dataset is relatively simple, and the improvement is not very significant. It’s unclear how much of the improvement arises from better prompt engineering versus structural innovations.
3. Baselines like ReAct and CoT do not require as much computational effort as HoT, and the comparison under different computing budgets lacks fairness. Although HoT claims to be efficient, the paper doesn’t provide detailed runtime or token cost analyses. The added multi-step reasoning (especially decomposition and path exploration) likely increases inference time, which isn’t quantitatively assessed.
4. There is a problem with the reference format in the paper.

**Questions:**

Refer to weakness.

---

### Official Review · Reviewer_eGHE · 2025-10-31

**Soundness:** 2
**Presentation:** 3
**Contribution:** 2
**Rating:** 2
**Confidence:** 5

**Summary:**

This paper introduces Holon-of-Thought (HoT), a structured, multi-stage prompting framework designed to improve the robustness and accuracy of LLM reasoning. HoT operates via a four-module pipeline: (1) Condition Extraction, to identify problem constraints; (2) Tree Explorer, to select an optimal solution path (out of N=3); (3) Adaptive Domain Decomposition, to decide if a problem needs division; and (4) Resolution and Aggregation, to solve the problem(s). The authors evaluate HoT on math and logic benchmarks, claiming SOTA accuracy over baselines like CoT-SC and AoT, and improved "robustness" (defined as lower accuracy variance).

**Strengths:**

Addressing the brittleness, error propagation, and condition-violation of LLM reasoning is a timely research direction. The 4-stage HoT prompt-engineering framework is well-described, intuitive, and clearly illustrated in Figures 1 and 2. The logic of first identifying constraints and then planning a solution path is well-motivated.

The ablation study (Table 3) provides clear evidence for the importance of Module 1 (Condition Extraction). The largest performance drop occurs when this module is removed, validating the paper's core premise. The paper deserves credit for attempting to measure robustness (Section 4.1.4, 4.2.3) beyond simple accuract. The appendix provides the full prompts, which is good for reproducibility.

**Weaknesses:**

A key weakness is the paper's limited novelty, as its architecture is largely derivative of existing, uncompared work. Module 2 (Tree Explorer) appears to be a simplified, fixed-branch (N=3) version of Tree of Thoughts (ToT), and Module 3 (Adaptive Decomposition) re-uses concepts from other decomposition methods like ThoT (Thread of Thought) and RDOLT(Recursive Decomposition of Logical Thoughts). The paper fails to cite or benchmark against these obvious predecessors, making it impossible to determine if this specific combination offers any new, meaningful contribution to the field.

The paper positions HoT as a structured reasoning framework but fails to compare against its most direct competitors. The current baselines (CoT-SC, AoT) are insufficient. This is the most significant flaw. Please evaluate HoT against Tree of Thoughts (ToT). The comparison should be normalized for cost (e.g., HoT with N=3 paths vs. ToT with a similar branching factor). Please evaluate against at least one other explicit decomposition method (e.g., ThoT or RDOLT) to validate Module 3.

The paper's strongest results (92.2% avg, Table 1) do not come from the HoT framework itself. They come from "HoT*," a 3-way ensemble of CoT, HoT, and a HoT-ablation (lines 365-367). This obscures the core contribution. Please clarify the contribution by removing the 'HoT*' result from the main comparison table. The paper's claims must rest on the performance of the HoT framework itself, not an ensemble.

All experiments seen to be run on N=200 samples, which is a tiny fraction of benchmarks like GSM8K. On N=200, the 1.0% gain of HoT over AoT (91.0 vs 90.0) is a difference of only 2 samples.

The authors explicitly state they "eliminate actual API calls or tool integration" and "simulate" actions in natural language. The core contribution of ReAct is the synergistic loop of generating an Action, having an external tool execute it, and receiving a new Observation. By removing this loop, the paper's baseline is just a Chain-of-Thought prompt formatted with "Thought" and "Action" labels, not a true Reasoning-Acting system. Please implement ReAct correctly (with a simple tool like a calculator for math problems)

**Questions:**

Please check weaknesses specified above.

---

### Note · Authors · 2025-12-25

I have read and agree with the venue's withdrawal policy on behalf of myself and my co-authors.